# Laughing about Religious Authority—But Not Too Loud

Lena Richter

Department of Islam Studies, Radboud University, 6525 HT Nijmegen, The Netherlands; l.richter@ftr.ru.nl

**Abstract:** In Facebook groups of young Moroccan non-believers, cartoons, memes, and jokes that mock religion are widely shared. By phrasing the messages in a humorous way, it is possible to express experiences and viewpoints that are more difficult to communicate in direct speech. Studying these forms of humor can reveal several themes, frames, and tropes that are important to many former Muslims, such as criticizing the legal restrictions of non-belief and countering stereotypes about non-believers. This leads to the following question: To what extent is humor being employed as an (online) tool for young Moroccan non-believers to challenge the religious status quo? To answer this question, the article analyzes numerous examples of religious-related humor during the COVID-19 pandemic and Ramadan. Hereby, it becomes clear that many jokes remain a limited and covert dissent strategy, as they are only shared among fellow non-believers. Yet, this article argues that jokes are an important method of differentiation, self-expression, and in-group identification that can build a fruitful ground for future activism.

**Keywords:** anthropology of non-religion; lived religion; online activism; humor; memes; Morocco

## 1. Introduction

*People keep asking me, 'Is COVID-19 REALLY that serious'? Listen y'all, the casinos and churches are closed. When heaven and hell agree on the same thing, it's probably pretty serious.* (Facebook post during the COVID-19 global pandemic 2020).

Looking at the numerous Facebook groups of young Moroccan non-believers, many aspects may catch one's attention: the sharing of personal stories, the quickly mobilized support if a group member gets in trouble, and the vivid discussions. Yet, one aspect, in particular, is difficult to overlook: the sheer amount of memes, cartoons, and jokes that flood the Facebook group wall. Humor, as the introductory quote illustrates, is one of the most popular ways of expressing experiences and thoughts among non-believers in different parts of the world. On the surface, Internet memes and other jokes might appear trivial. While they often appear to lack seriousness, they are an intrinsic part of today's digital culture (Milner 2012; Shifman 2013) and carry important social, emotional, cultural, and political messages (Miltner 2018; Bennett and Segerberg 2012). Beyond that, memes can be an important part of lived (non)religion, as they are a common and participatory expression of meaning-making in everyday life (Aguilar et al. 2017).

Despite the long-established use of humor as a political tool (Anagondahalli and Khamis 2014), the study of both online and offline activism has rather focused on serious and structural aspects, such as leadership styles, strategies, and mobilization (Hiller 1983). The lighter and creative sides of (online) activism, such as rap (Gruber 2018), art (Horváth and Bakó 2016), or humor (Hiller 1983), have been often neglected. By looking at the use of mockery among young Moroccan non-believers, this article aims to shift the focus to the humorous side of online activism. During numerous interviews, which I conducted between 2016 and 2020 about the experiences of being not religious, the role of humor was often mentioned. The gender-mixed interview group encompassed Moroccan non-

believers[1] with a Sunni Muslim background, who were between 18 and 35 years old, mostly lived in Casablanca or Rabat, and belonged to the educated middle class.[2] Based on the recommendations of my interviewees, I selected four different Facebook groups[3] to explore the role of humor further:

1. Atheists in Morocco is a private hidden group with around 4000, mostly young, members from Morocco and the diaspora, that mainly interact in English. This group has been the main source of jokes for this analysis.

2. Marocains pour la Laïcité is a francophone, private, and visible group that includes 16500 members with different (non)religious viewpoints, who advocate for the separation of religion and politics in Morocco.

3. MALI مالي؟ is a private-visible group with 7000 members that belongs to the public page of MALI (Mouvement Alternatif pour les Libertés Individuelles). Its activist content is shared in different languages, such as French, English, and Arabic.

4. Ramadan for everyone is a small, private, and hidden group that has been mainly active in 2018. It is meant for people who do not fast, such as non-believers, non-fasting Muslims, or Christian converts.

While these groups have different aims, such as exchanging ideas or organizing meetups, this article focuses on jokes as a potential tool for activism. In this context, activism can be defined as contesting hegemonic religious power structures. Recognizing that jokes might be shared with a non-activist purpose or that non-believers might prefer other forms of activism or not to engage in activism at all, this article narrows the focus on non-believers who see jokes as (part of their) activism. This article thus takes a look at a specific group of people (mainly young, urban, educated, and middle-class Moroccan non-believers) engaging in a concrete practice (sharing religious-related jokes online) for a particular purpose (challenging religious authority). Hereby, religious authority is understood by most interviewees as being an omnipresent structure that permeates the whole Moroccan society, rather than only being incorporated by individuals or religious scripts. Against this backdrop, the following question arises: To what extent is humor being employed as an (online) tool for young Moroccan non-believers to challenge the religious status quo?

To answer this question, the article starts by analyzing the legal and political space for Moroccan non-believers, especially for those who aspire to engage in (online) activism. Following the contextualization, I describe the forms, tropes, and dominant themes of humor, such as legal restrictions, mocking religion, and reflecting on society's views towards non-believers. First and foremost, I refer to religion-related jokes made during the COVID-19 pandemic, when activists relied even more on online activism, and during Ramadan, which can be seen as the high season of (humorous) activism. After this thematic analysis, I underpin these research insights with the theoretical functions and strategies of using humor as an activist tool. While the focus is put on how humor can (in)directly challenge religious authority, humor can also fulfill other purposes. It can contribute to bridging (non)religious disagreement, to create identification among non-believers, and to establish differentiation towards the religious majority. Finally, I discuss the limits of humor as an activist tool, such as its potential to create division and to remain a covert dissent strategy.

## 2. Spaces for (Online) Activism in Morocco

In Morocco, activism that openly advocates for freedom of conscience is rather restricted, as many activists fear the legal and social consequences. Suffering from stigmatization, activists reported cases of (verbal) violence by family members, investigations by

---

[1] Acknowledging that the self-identifications of this group can reach from being atheist to agnostic to cultural Muslim, I opt for the broad term "non-believers" (*la dini*).

[2] Names of people are pseudonymized, unlike otherwise wished by the interviewees.

[3] Names of hidden groups are pseudonymized.

authorities, and obstacles in professional, educational, and private life. As a consequence, some prominent voices, such as Kacem El Ghazzali, had to flee the country.[4] The restrictive situation for non-believers becomes visible in the Freedom of Thought ranking (2019), where Morocco takes place as 182 out of 196.[5] While the Western frame of this ranking needs to be taken critically, it gives an indication as to why many non-believers prefer not to engage in activism or opt for more indirect forms of activism such as humor. Moreover, the room for humorous activism is influenced by the semi-authoritarian Moroccan context, which restricts some freedoms but offers others (Ottaway 2003). Both citizens and politicians can use humor, but political satire is especially seen as a weapon of the weak and those in opposition (Nilsen 1990).

The Moroccan government has often been applauded for taking a relatively liberal hands-off approach. Yet, in the current post-Arab uprisings environment, surveillance and censorship did not vanish, but only became more advanced (Iddins 2020). For instance, by combining new spyware with traditional ways of intimidation, such as phone tapping or spreading false rumors. At the same time, ways to circumvent restrictions developed (Shayan 2016). Humor tests the border of what is still tolerated to express. On the one hand, some jokes enjoy a free pass as they are "not meant seriously" and provide a space of liberty that allows people to vent frustrations (Davies 2007). On the other hand, some topics, that touch the troika of "allah, al-watan, al-malik"[6] (Kettioui 2020) cross that line and are labeled as blasphemous. The awareness that it is not possible to criticize or joke about certain topics leads to (self-)censorship (Rahman 2012) and a try-and-see ethos that tests the limits of freedom of speech (Iddins 2020).

The room for humor is also influenced by the religious or secular morality the state and society base their identity on. In the case of Morocco, morality is closely linked to Islamic values. This is mirrored in the preamble of the constitution which defines Morocco as a "sovereign Muslim State, attached to [ . . . ] its indivisible national identity. Its unity is forged by the convergence of its Arab-Islamist, Berber [amazighe] and Saharan-Hassanic components." Islam and humor do not need to oppose each other. On the contrary, humor is an important part of Moroccan society. Yet, joking can undermine the Islamic authority which is often associated with discipline and modesty (Bayat 2007). Most importantly, Moroccan religious and political authority are closely tight, as the King is not only the political head but also the Commander of the Faithful.[7] This title gives him much symbolic importance and power, such as the legitimacy to preside over the High Council of the *Ulemas*[8], which is the only instance entitled to issue *fatwas*[9]. His position is backed by the constitution which specifies that it is not allowed to criticize Islam or the monarchy (§ 175). In this respect, questioning religion, with or without jokes, becomes a problem, because it indirectly undermines the authority of the King.

While the current King, Mohammed VI enjoys a liberal reputation and promotes religious coexistence, especially with Jewish co-citizens (Maghraoui 2009), he remains cautious not to weaken the monarchy's religious basis (Benchemsi 2015). Therefore, the King encourages a certain form of Islam that gives him control and limits other forms of religion that might question his power.[10] While many NGOs support the idea of individual freedoms, due to the sensitive nature of the topic, not many civil society players (can) work openly on the right not to believe. However, as part of the Arab uprisings, the February 20

---

4　Interview Kacem El Ghazzali, 7 December 2016, Skype.

5　The Freedom of Thought Report is conducted by Humanist International and measures the following categories: (1) Constitution and Government, (2) Education and children's rights, (3) Society, Community, and Family, and (4) Freedom of expression, advocacy, and humanist values. Humanist International was founded in 1952 in Amsterdam and has its headquarter in London.

6　God, the country, and the King.

7　Amir al-Mu'minin.

8　Islamic scholars.

9　Religious consultations.

10　For instance, the movement Al Adl Wa Al Ihssane, which does not recognize the religious legitimacy of the King.

movement raised more attention on the topic (Thompson 2015). Without embracing a too utopian idea of technological determinism, the Internet became an open and connecting space for activists and critical journalists of whom a significant part identified as not religious (Iddins 2020). In this context and its aftermath, the new generation has shown a strong proclivity for political satire, cartoons, and mockery (Kettioui 2020; Harutyunyan 2012). For instance, in Facebook groups, such as "Mohammed VI, My Liberty is More Sacred Than You." With the visual rush, artistic critique and a call for more civil rights and freedom became visible on social media, television, and the streets (Khatib 2013; Baylocq and Granci 2012). To calm the mixed calls for democratization and secularization, the King proposed a reformed, albeit ambiguous, constitution in 2011, that name-checked the demands of Moroccan liberals and conservatives, as well as international institutions (Benchemsi 2012).

In general, this new constitution promotes both Islam (§ 1 and 3) and the freedom of belief (§ 3 and 25). At the same time, Morocco signed several UN treaties concerning freedom of religion, conscience, and thought, such as the Resolution on the Freedom of Religion or Belief (UN Human Rights Council 2013). The government only commits itself to these international conventions as long as it is compatible with the national identity based on Islam (preamble). Albeit important reforms, the family code *Moudawana* and the penal code still restrict religious liberties, including the right not to believe. Several Articles of the Penal Code prohibit anyone from affirming any views other than those of Islam (Benchemsi 2012). Among others, Article 220 of the Penal Code criminalizes "shaking the faith of a Muslim", which makes it difficult to utter bold jokes about religion. The disrespect of religious practices, blasphemy, and the violation of public morality and virtue are strictly forbidden (§ 483). For example, recently actor Rafik Boubker got accused of blasphemy when he praised the benefits of alcohol for connecting with God. He risked being sentenced to up to two years of imprisonment and a fine of up to EUR 20,000. Besides, conversion from Islam, missionary activities, and the distribution of non-Islamic religious material are criminalized (Benchemsi 2015). For this reason, Facebook administrators of groups such as "Atheists in Morocco", often do not let Muslims enter the group, to avoid being accused of dissuading Muslims from their beliefs.[11]

Despite the restricted room for humor, it is still one of the most popular forms of online activism among Moroccan non-believers. In general, non-believers are part of the young and social-media generation that has moved their activism to the Internet as they mostly favor personalized and issue-specific cyber-activism instead of institutional politics (Zerhouni and Akesbi 2016). Like other forms of online activism, the use of humor among young Moroccan non-believers has developed from the first activists who began with blogging, to switching to Facebook and other social networking sites. By now, many Moroccan "gen Y-ers" advocate for individual liberties online, even at the risk of offending or upsetting the religious status quo that underlines the political system (Rahman 2012). The Internet has especially given a counter-voice to non-believers and other minority groups that are often not fully represented (Mohammed 2019). Social media platforms provide a relatively safe space of resistance and consciousness-raising in comparison to more overt forms of activism. On the Internet, topics are being discussed in alternative ways to the ascribed social norms (Fileborn 2014). Thus, while Moroccan non-believers can exchange their jokes about religion on Facebook, they cannot do it in the same way outside of social media.

The degree to which Moroccan non-believers can and want to make jokes, both in public and online, also differs from person to person. While most of the active Facebook group members merely circulate, like, or comment on jokes, others also engage as pro-sumers (Ahluwalia and Miller 2014) by (re)making, (re)adapting, or (re)mixing memes (Shifman 2013). In particular, closeted non-believers might not want to make certain jokes in public, as that would indicate their not religious or even anti-religious viewpoints. Due

---

[11] Interview with Facebook administrator "Atheism in Morocco", 24 December 2020, Facebook Messenger Call.

to the controversial nature and social taboo that non-believers face, joke-tellers might be afraid of the reactions of Muslim family members, friends, and colleagues. Hereby, one's position in terms of, for instance, gender, safety net, living abroad or in Morocco, and socio-economic situation influences the ability to tell a joke in public. This is added to the personal motivations not to openly tell jokes, for example, due to not wanting to offend religious co-citizens. Considering these different aspects, it is important to keep an intersectional perspective in mind: who can make what kind of jokes in which context and in front of which audience?

Essentially, the Moroccan context leaves not much space to openly mock religious authority, as it is closely linked to political power. Consequently, jokes are mainly made in the private hidden online sphere among like-minded non-believers or Muslims who share similar views, but are rarely shared in public. Exceptions are a few activist groups, such as the MALI movement, that does not only share, but also creates cartoons. The dissatisfaction with the legal situation in addition to societal taboos of being less or not religious also becomes the subject of jokes itself, which I turn to in the next section.

### 3. Joking about Religion: From Beers in Ramadan to Prayers against the Coronavirus

Humor that mocks religion as the foundation of political power can take different forms, such as puns, rhetoric questions, mockery, witticisms, anecdotes, caricatures, or satire (Schutz 1977). It can be a part of face-to-face communication but also mediated via newspapers, television, and social media (Driessen 2015). As bottom-up and low-key resources, memes are especially popular among non-believers (Nissenbaum and Shifman 2017). Ironically, the term meme was first introduced by new atheism thinker, Dawkins (1976), in his biological research. Since then it came to be understood as a common tool of creative expression of today's digital culture (Aguilar et al. 2017).

Jokes enjoy longevity and transnationality, as they change forms according to the religious and national context while keeping the same core (Anagondahalli and Khamis 2014). Although almost all Facebook group members grew up in a Muslim environment, they often take jokes from bigger atheist or agnostic pages which are frequently coined by a focus on Christianity. This became already clear with the introductory quote that referred to closed churches instead of mosques. Sometimes jokes are also adapted to the Islamic context by changing the name of the religious leader or prophet. For instance, one cartoon shows an empty list with the caption "A list of things that God has done during the pandemic". The same picture is shared and slightly adapted to "A list of things that Allah has done during the pandemic".

While some non-believers do forward the jokes to friends in- and outside Morocco or aim to openly address those in power, most non-believers tell their jokes only in private and/or hidden Facebook groups. Especially, bold jokes remain only internally communicated. Many non-believers share the view that, for now, it is still too much of a risk to share these jokes with a bigger audience. Even in private and hidden groups, some non-believers do not feel completely free to express their opinions, out of fear that screenshots of the group members and their conversations may be taken and exposed beyond the Facebook group. Others refrain from joking about religious authority, as they consider it irreverent or not funny. Despite that, jokes have the potential to travel very quickly, intended or not, beyond the initial network of circulation. Most of the jokes are told in colloquial English or French, but if the jokes are translated into Darija, which has become more popular since the February 20 movement, an even wider group is reached (Kettioui 2020). Moreover, the development of making jokes more visual and shorter, such as in the form of tweetable one-liners, has contributed to a possible widespread and quick outreach.

The form, language, and content of the jokes also depend on the Facebook group. In the hidden group Atheists in Morocco, memes are especially popular among the predominantly young and English-speaking members. Jokes cover a wide range of subjects, such as the existence of Allah or religious followers. The shared memes are often relatively bold and sometimes involve sexual innuendos. In the visible group Marocains pour la Laïcité, mainly

French cartoons that address subjects such as the headscarf debate and jihadism are posted. In the MALI group, many political cartoons are shared which, among others, focus on the legal situation of non-believers in Morocco. The political messages of the cartoons reach a large audience, as the jokes are not only translated into different languages, but also made both in public and in private. In the hidden group Ramadan for everyone, mainly jokes in Arabic related to fasting are posted.

As humor is a possibility to express experiences and opinions, jokes can give an indication of which topics are important to non-believers. Looking at the four Facebook groups, I identified recurring themes of humor which can be broadly divided into (1) criticizing the legal situation, (2) mocking religion, and (3) reflecting on society's views towards non-believers. Firstly, humor addressing the legal situation might be the most political category of the three, as it directly criticizes the restrictions related to freedom of conscience. Especially, laws, which have been partly drafted by French colonial administrators who stayed after independence, are in the spotlight (Zirari 2016). One of these laws is Article 222, which states that it is not possible to eat on the street during Ramadan. Based on this article, the police arrest or fine people each year who break the fast in public. As everyone had to stay (and eat) inside due to the pandemic, activists ironically celebrated 2020 as the first year with (almost) no arrests. For instance, Figure 1 shows a disenchanted police officer who is asking: "but boss with the lockdown what am I gonna do with Article 222?". A similar cartoon portrays a man calling the police asking, "My neighbor is drinking water in his living room, can I beat him up or are you gonna arrest him arbitrarily?". Most jokes are slightly exaggerated as the police usually do not apply the law very strictly, especially if someone would eat without causing attention. According to Soufiane, a young web designer, the hyperbole is used on purpose to show the disproportion of being arrested or having to pay a fine for a basic need, such as drinking water.[12] Jokes shared among non-believers do not only address their own restrictions, but also the legal struggles of perceived allies. For instance, the LGBTQ+ community, religious minorities, and feminist groups that also fight for more rights and liberties. In relation to that, some jokes criticize the considered religious basis of criminalizing, for instance, abortion (§ 453) or homosexuality (§ 489). In this context, joking statements such as "Allah is gay" or a *Kaaba*[13] in rainbow colors are common.

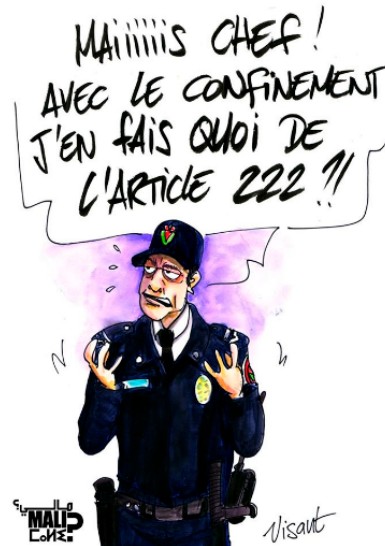

**Figure 1.** © MALI/Visant.

---

[12] Interview Soufiane, 18 December 2016, Rabat.

[13] A sacred site in Islam that indicates the direction of prayer and is the destination of the pilgrimage *hajj*.

Secondly, jokes can target religion as a belief system, and mock its religious followers, leaders, and figures. In this regard, one of the running gags is to ironically apply religious terminology for mundane matters. For instance, by saying "Oh my God, I worship that" or "Thank God I'm an atheist". Not only religious language but also behavior is sometimes imitated. For instance, one time I was invited to join a group of activist non-believers for a drink. When the sound of the *adhan*[14] filled the air, one of the activists jumped up from her chair, loosely wrapped a scarf around her hair, and said jokingly "bye, bye I'm going to pray". Other jokes repetitively reproduce religious scenes, for instance between the prophet Muhammad and his first wife Khadija. Humor that directly addresses religion itself often mocks the lack of religious evidence and contrasts religion with science. In many jokes it is especially criticized that religion often gets the credit for scientific achievements. For example, when a doctor successfully finishes a surgery but the person only thanks God/Allah. Sometimes, this kind of humor can be quite dark, for instance, when a picture with a boy having cancer is titled "God works in mysterious ways". Other popular examples that aim to underline the perceived illogical character of religion (see Figure 2), are the biological impossibility of Noah's ark or the portrayal of Jesus as the only white man in the Middle East. Contrasting religion and science became especially visible during the COVID-19 pandemic, when religion was portrayed as helpless and science as a rational solution. In addition to that, many jokes make fun of the claim of each religion to be the only religion: "There are 5000 gods, but yours is the true one". Many cartoons also joke about "religious plagiarism". For instance, the "pupils" Islam, Christianity, and Judaism sit in a classroom and copy from each other. Another repetitive joke is to quote phrases from the Quran which the joke-teller perceives as illogical or violent. In this respect, non-believers often joke that they know the Islamic scripts better than believers do, or that what made them leave Islam was actually reading the Quran.

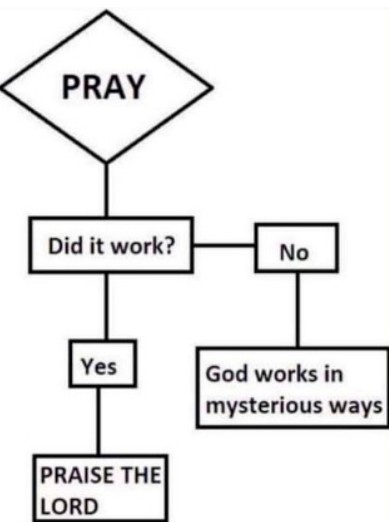

**Figure 2.** © unknown.

Furthermore, jokes that portray religion as misogynistic are common. Especially popular are cartoons of women wearing the burqa. In Figure 3, one woman is commenting on another woman who "dares" to show her ankles. Moreover, beauty contests where everyone looks the same, and bus seats or bin bags that are "accidentally" taken for women with burqas are frequently made jokes. This is often contrasted with men wearing fewer clothes: a woman, wearing a burqa, is sweating on the beach, while her husband next to her only wears swimming trunks. Another target are Muslim feminists, in jokes that portray feminism and Islam as incompatible. Besides that, in many memes, Islam and other

---

[14]   The Islamic call to prayer recited by a *muezzin*.

religions are often described as violent, which is contrasted with the self-display of being religions of peace. For example, aliens look down on the planet Earth and comment, "they are fighting over which religion is the most peaceful". This leads to a picture of religion in general and Islam in particular as being irrational, misogynist, and violent. Yet, these humorous presentations are not uncontested among non-believers.

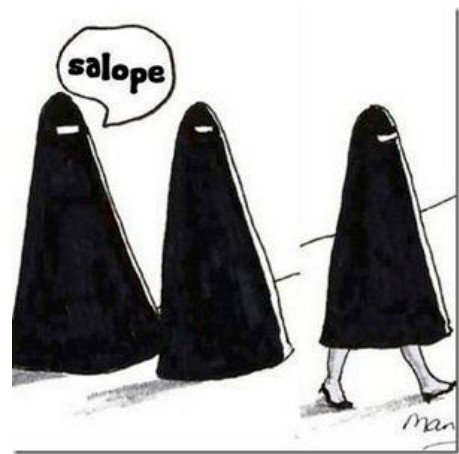

**Figure 3.** © man.

When jokes target religious communities, they are often depicted as blindly following their beliefs. This is, for instance, visualized by showing a scan of their head, which reveals emptiness instead of a brain. This also became evident during the COVID-19 pandemic, in jokes such as "praying only helps if you add soap". Portraying religious followers as ignorant is also the aim of mocking the arguments of believers, for example, by sharing screenshots of discussions with believers about the existence of Allah. Religious people are sometimes also presented as insanely talking to an imaginary friend. In one cartoon of the American sitcom *The Simpsons*, Marge talks to her psychologist and says "Oh, I was just praying to God that you will find me sane" and he answers "I see. And this God is he in the room right now?" In a similar dialogue, someone is asking: "Do you have any cases of mental illness in your family" and the interlocutor replies, "I have an uncle who believes in God". When it comes to religious followers, especially fundamentalists, portrayed with long beards and weapons are targeted. Most non-believers are more hesitant to make fun of "ordinary" Muslims as they often have good relationships with their religious co-citizens.

Furthermore, religious leaders are frequently the focus of humor. For example, in the following joke: "A priest, a rabbi, and an imam walk into a bar. The bartender says: what is this? Some kind of a joke? Muslims don't drink alcohol". During the COVID-19 pandemic, the number of jokes about religious leaders increased (see Figure 4). In many cartoons, they were begging scientists to quickly develop a vaccine to calm down their religious community. Moreover, rich sheiks or televangelists who ask for private jets are mocked. Religious leaders in higher positions, such as the pope, receive their share of mockery as well. In comparison to the numerous jokes of pedophile priests, imams get off quite well and are only scarcely addressed. This shows again that a lot of the content of the jokes is not created in Morocco, or other countries with a Muslim majority population, but is derived from Western sources.

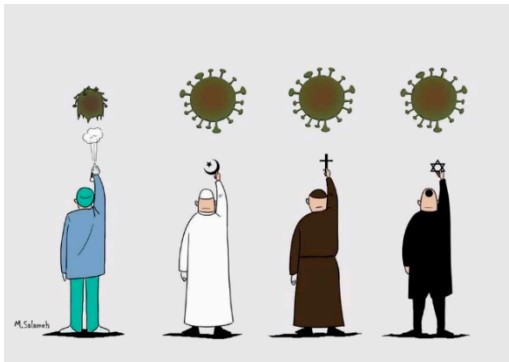

**Figure 4.** © Salameh.

Besides religious leaders, prophets are frequently portrayed tropes. While there are also many puns about prophet Muhammad, such as "atheism is a non-prophet organization", there are even more jokes about Jesus (Aguilar et al. 2017). Apart from the already mentioned relation to Western satire, this might be explained by the taboo concerning visualizations in Islam and the sacred character of the prophet Muhammad. The favorite protagonist of jokes seems to be God's or Allah's opponent: Satan. In relation to the COVID-19 pandemic, one meme shows Satan making an astonishing face which is titled "Satan looking at God's plan for 2020". Another meme of Satan says that plagues, great floods, and pandemics are God's department and a list compares God and Satan, where God checks the following boxes: (1) committed genocides, (2) asked a man to kill his only child, (3) allowed slavery, (4) made women inferior to men, and (5) caused all the suffering in the world. In this light, the list of Satan seems small: (1) convinced two people to eat fruit. The roles become reversed as Satan becomes the "good guy", while God is portrayed as the bad one. References to Satan and hell are equally frequent in non-pandemic times. From time to time, a picture of Jesus looking around the corner is posted in the Facebook group Atheists in Morocco, saying: "I'm here to remind you that you will all go to hell" and statements that hell must be more fun are common.

Thirdly, jokes can reflect on the personal situations and everyday concerns of non-believers. Hereby, jokes often embrace self-satire, a form of visual critique which is flourishing. This form of humor is used especially during Ramadan, such as stating: "No offense but pretending to be fasting is actually harder than fasting". In other jokes, non-believers describe themselves as fighters who have to survive the "hunger games"[15] and desperately look forward to the end of Ramadan, which is, for instance, symbolized by a beer bottle next to the crescent (see Figure 5). Some self-mocking jokes respond to clichés of being immoral, such as "Yes, we eat babies for breakfast". Figure 6 shows how the perception of the same person can change when he is saying that he is not believing in God. From a "normal person", he is changing into a Satanist with long black hair praying to the Sigil of Baphomet.[16] These and other jokes very much relate to the experiences of non-believers. During an interview, Soufiane told me: "People at school talk about me. They say I worship the devil. I don't feel safe".[17] At the same time, the self-portrayal can also be very positive, as non-believers mostly come off quite well. For instance, there are even t-shirts with the slogan "Support intelligence- sleep with an atheist". This example also shows that many jokes can be quite sexual in nature, which touches on multiple taboos at the same time.

---

[15]  Reference to the film and book series Hunger Games.

[16]  The pentagram of the Church of Satan.

[17]  Interview Soufiane, 18 December 2016, Rabat.

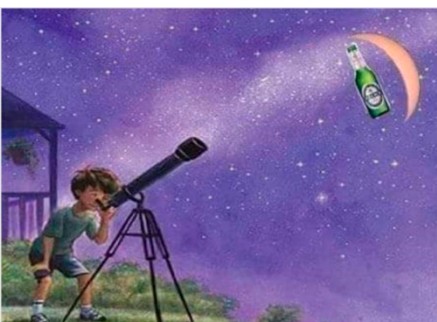

**Figure 5.** © unknown.

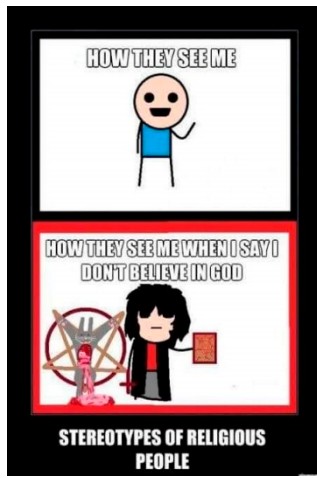

**Figure 6.** © unknown.

Other jokes focus on the personal restrictions that non-believers encounter. Figure 7 shows a mother holding the mouth of the "dog" (her child) to not say anything negative or critical about religion in the presence of the rest of the family. In this and other cartoons, social mechanisms such as *hchouma*, which can be roughly translated as "bringing shame" to oneself or the family, become evident. Furthermore, how to behave correctly during Ramadan has been a topic in jokes for a long time. This can be exemplified by comparing the following two cartoons (Figures 8 and 9), which have been published with around twenty years of discrepancy.[18] The first cartoon poses the question: "You eat during the middle of the day?", upon which the old man answers: "It's chewing gum". The second cartoon portrays two police officers who comment on a man walking down the street: "He's smiling during Ramadan- that's suspicious".

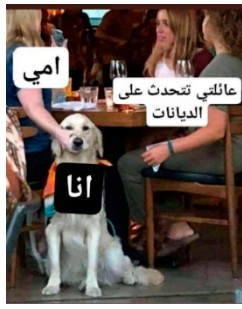

**Figure 7.** © unknown.

---

[18] Figure 8 was published in the daily newspaper L'Opinion, and has been illustrated in the dissertation "Fasting and Feasting in Morocco: an ethnographic study of the month of Ramadan" (Buitelaar 1991).

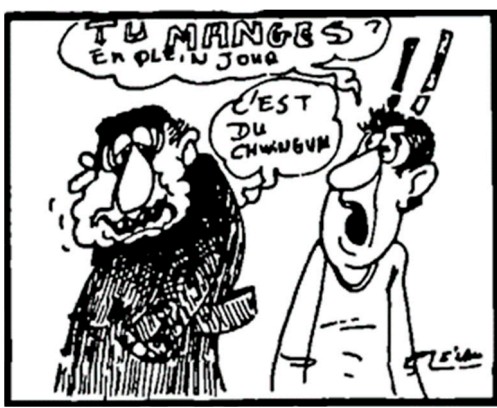

**Figure 8.** © L'Opinion.

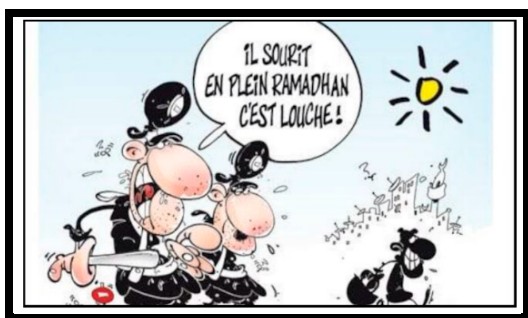

**Figure 9.** © Dieu, journal Liberté Algérie.

Of course, many members of Facebook groups also make jokes that are not (directly) related to religion, such as memes about former US President Donald Trump. However, a specific kind of humor can be distinguished that criticizes laws based on religious norms, mocks religion, as well as its followers, leaders, and figures, and reflects on clichés and stereotypes about non-believers. This leads to the question: what is the purpose behind these different kinds of jokes? Can we see them as forms of activism or do they have a mere entertaining purpose?

### 4. Humor = An Activist Tool?

As Aguilar et al. (2017) have pointed out, religion-related joke-tellers can apply different frames. Memes about religion do not need to be critical or negative, but can also be positively employed by religious followers, for instance, when the promoting religion frame or the playful frame are used. Yet, in the examples discussed above, especially the questioning frame and the mocking religion frame are employed (ibid). The questioning frame explores common and often negative generalizations to raise objections or doubts toward perceived religious contradictions. This frame was visible in jokes about religion, including their textual basis, promoters, and followers. By doing so, this frame aims to encourage debates and exchange. The mocking religion frame goes a bit further in its critique of religion and aims to undermine beliefs. Applying these frames facilitates different purposes.

Returning to the main question of this article—humor can indeed have a subversive purpose that targets those in power, and therefore plays a key role in challenging the power equilibrium. How religion is presented in these counternarratives can challenge "traditional" forms of belief and undermines religious institutional structures (Aguilar et al. 2017; Kettioui 2020). According to the punch theory, traditional forms of satire in general and political satire in the Arab world, in particular, have been nearly always directed to punch upward, at figures of authority rather than downwards (Mulder 2018). Thus, usually, satire is a critique of the powerful made by the disempowered, which offers a

chance for dialogue (Gruber 2018). This is also the case of Moroccan non-believers who find themselves in a lower power position and criticize the hegemonic religious authority. Jokes can reverse the status of people by portraying inferiors as superiors and vice versa (Anagondahalli and Khamis 2014). Yet, power constellations are more complex and go beyond being religious or not. Most non-believers also find themselves belonging to the educated middle- or even upper class. In this case, making jokes about religious people that play with stereotypes about the working class, such as ignorance or superstition, can be a form of punching downwards.

When it comes to humor as a tool to challenge the religious status quo, either the mere fact of joking or the content of the joke can be subversive. Often, jokes carry messages that directly criticize the legal or political situation for non-believers. Consequently, humor can aim for the enforcement of certain norms and rules in order to change a situation. It can also serve as a means of clarification to explain the communicator's viewpoint on religion (Meyer 2000). In this respect, humor can also point out the taboo (Kettioui 2020) around leaving Islam. Memes, and cartoons that aim at challenging the religious status quo, are designed to get attention, and in doing so, to provoke and shock (Mulder 2018). Challenging authority can also be indirect, by using self-satire. In a Muslim-majority context where agency for non-believers is rather limited), the purpose of ridiculing oneself is not to create victimhood but empowerment (Anderson 2013). By visualizing their struggles and restrictions, Moroccan non-believers create a subtle critique that underlines the perceived absurdity of the situation and goes beyond the direct mockery of religious authority (Mulder 2018).

The provoking nature of many jokes can also have the purpose of differentiating oneself from the religious majority. The differentiation became especially clear by looking at jokes made during the COVID-19 pandemic. Many non-believers shared memes that mocked the behavior of religious followers and leaders to the pandemic and compared these reactions to the ones of scientists and doctors which were depicted as fact-based and rational. Praying was portrayed as irrational and ineffective and religious leaders were shown as being dependent on scientists. This led to the construction of an opposition between two groups that either opt for science or religion without leaving space for in-between positions. Furthermore, other dichotomous notions of modern vs. backward, violent vs. peaceful, and misogynic vs. feministic were being reinforced. When believers and non-believers are being contrasted, the differentiation might lead to a form of moral superiority by putting the freedom not to believe above religious views (Anagondahalli and Khamis 2014).

While humor addressing Islam can be polarizing (Molokotos-Liederman 2019), jokes can also bring Moroccans with different ideas together and bridge incongruities. Self-mockery or an ironically used "*inshallah*" (or "*outshallah*") are common among both non-believers and Muslims.[19] For instance, during the COVID-19 pandemic, people with different (non)religious ideas made fun of the government's promise—*inshallah*—of a short lockdown. Humor does not always need to be "laughing about somebody" but can also be a "laughing with". In addition to that, most jokes are less intended to upset fellow citizens, but instead aim to contest the hegemonic religious-political power that denies non-believers equal rights. As medicine student Anas says: "I also have a lot of friends who are not atheists. But they are very open. We make fun of each other!"[20] By doing so, laughing can help to diffuse tension, disagreement, and dissonance (Gruner 2017). To avoid conflict, it can also be a way to convey a message that cannot be said directly (Anagondahalli and Khamis 2014). At the same time, humor about religion also brings together non-believers from different religious backgrounds. While some groups, such as "Atheists in Morocco"

---

19  While I speak here of two groups (non-believers and Muslims), I would like to stress that religious identifications are more complex and fluid and go beyond this binary division.

20  Interview, Anas, 6 December 2016, Temara.

mainly include non-believers who grew up in a Muslim environment; other, international pages, such as Atheist Republic, attract people from all over the world.

Humor can also attempt to create identification, as it connects the joke-teller to the audience (Meyer 2000). By doing so, jokes help to identify common struggles, dilemmas, and shared values (Driessen 2015). This gives a human face to non-believers who deal with different stereotypes and prejudices, such as being perceived as immoral. Revealing one's personal situation, and therefore also one's restrictive environment, makes the audience a fellow witness, which aims at creating awareness, solidarity, and understanding (Mulder 2018). Thus, the personal component about being not accepted within society or one's family, provokes empathy with the joke-teller. This is especially the case during Ramadan, when many non-believers feel restricted to eat in public and, depending on their living situation, also in private. Jokes about the common experiences and struggles weld together non-fasters and create a group feeling that strengthens the emotional ties and norms. Being able to understand the cultural code behind memes and other jokes that mock religion and finding its content funny can symbolize belonging towards other non-believers (Nissenbaum and Shifman 2017). Consequently, joking about a common target increases the internal group feeling, in-group trust, and cohesion (Gruber 2018). However, many nonbelievers also disagree with jokes that portray religion as unscientific, misogynistic, and violent. The different understandings of humor become clear in the vivid discussions in the comment section that follow the post of a controversial joke. In some cases, the disagreements of what is considered funny make group members leave the group.

In short, humor can perform and combine different purposes, such as challenging authority and bridging incongruities. It can also mark group boundaries, by creating a feeling of identification and belonging to (digital) communities. At the same time, this can lead to differentiation, as it can also be a tool to test and challenge social cohesion (Driessen 2015). While not every joke might directly challenge religious authority, most jokes contribute indirectly to questioning authority. Expressing agency, in-group identification, and outgroup-differentiation can be a fruitful ground for future activism. Finally, the purpose and intention of the jokes also depend on the group and the person that makes or shares the joke.

## 5. The Limits of Humor as an Activist Tool

As we have seen so far, humor can be a way for many activists to challenge religious authority but not every non-believer that shares jokes about religion has automatically an activist agenda. Many prefer not to engage in any kind of activism, for instance, because they focus on their private life or consider other topics more important. Others do engage in activism related to the rights of non-believers, but opt for other forms of activism, such as seeking dialogue, trying to change religious aspects of the school curriculum, or taking part in demonstrations. Moreover, those who see humor as an activist's tool recognize its limits. Most humorous messages only reach a small group. Among others, language skills which can range from English and French to Darija and Tamazight, depending on the group, are required. Remarkably, only a few jokes are made in Fusha—the official language of the political and religious elite (Kettioui 2020). Besides language skills, other necessary pre-conditions are (Internet) literacy and Facebook access. In Morocco, 60% of the population has access to the Internet and Facebook is one of the most frequently searched and visited sites, especially among young Moroccans between 18 and 35 (Rahman 2012). Mouhcine, a comedian and founder of the Facebook group "Ramadan for everyone" suggests that in order to sensitize more people for (non)religious minorities, it would be good to share funny videos also outside of social media. For him, also, non-digital alternatives, such as stand-up comedy might be a good option.[21]

While in theory, around 12 million Moroccan Facebook users do have access to Facebook groups, in practice, the interaction remains mainly restricted to non-believers and

---

[21]　Interview Mouhcine, 2 June 2019, Casablanca.

does not involve Moroccans with more religious viewpoints. The degree of intermingling also depends on the nature of the group. Some group guidelines explicitly state that Muslims are not allowed to enter, as the group is "intended to be a safe space for Moroccan non-believers." Such Facebook groups protect themselves by being not findable purely based on the group name. New members need to be invited and are first checked by the administrators if they do not pose any threat to the group. Additionally, entering these groups works like a self-filter: mainly those interested in the topic would know about these pages or would want to become members of the Facebook groups in question. As most members feel part of a trusted community of like-minded people, they communicate intimate opinions, thoughts, experiences, and jokes. Omar, an engineer whose parents are non-believers as well, describes the group as the following: "It's really enjoyable. It's good that there are no Muslims in it. In the group, I can make jokes about a lot of things."[22] Other groups that have a more secular character are open for Muslims as well, and groups such as "Ramadan for everyone" might also attract Moroccan Christians and other groups that do not fast.

On the bottom line, the jokes remain in the private or hidden sphere, out of sight from the views of relatives, curious neighbors, and colleagues. This kind of covert dissent (Anagondahalli and Khamis 2014) is more frequent when there is acceptance of, or resignation to, the status quo. This might be the case for most Moroccan non-believers who are aware of the societal taboo around leaving Islam and might not see it feasible (or desirable) to change that from one day to the other. Exceptions are activist organizations such as the MALI movement or the international page Atheist Republic who do take the risk to tell jokes in public because they consider it necessary to break taboos and to speak up. In this respect, the more people tell a certain joke, the more it gets normalized, and the more the shared accountability is spread. Thus, it might become possible to make the same jokes in public, which are now merely told in private.

Due to the covert nature of making jokes about religion in the web 2.0 (O'reilly 2009), this kind of activism has been perceived as rather safe and passive. According to Hicham, a 21-year-old student, online activism alone is not sufficient: "We just hear their voices on social media and social media is not enough. People are afraid, that's why they don't go on the street. People are afraid of the reactions, are afraid to get beaten up, afraid of getting arrested. If all atheists would go on the street it would change a lot. People would know that we're a lot of people, that we're a mass. People just need to say: today we will do it. Today we will do the sitting but today I'm wondering too if I should go on the street."[23] Thus, taking part in street activism can have far-reaching consequences, such as losing one's job, putting oneself and family members at risk, and jeopardizing relationships. However, relying only on online activism also risks being derided as "slacktivism", as relatively little commitment is needed for posting or sharing some scattered jokes on Facebook (Shayan 2016).

While online activism might be a safer place for minority groups in comparison to street activism, perceiving Facebook as a private and shielded sphere can also become problematic. The Internet still bears several risks, such as online harassment, digital public shaming, trolling, and surveillance (Shayan 2016). Therefore, some non-believers prefer to use fake Facebook profiles or names. Although the Internet offers some anonymity, the peril remains of being arrested, identified, or intimidated both by insiders of the group and outsiders. This has been the case for Soufiane when he was 18 years old: "I posted stuff about Islam on Facebook because that's the only place I feel safe to share my thoughts, but I forgot about the fact that I have also Muslim family members and friends on Facebook. My aunt saw my posts and told my mother about it. This caused a lot of problems. We were fighting. My grandma left us- she moved out. My mum wanted to move out too. So I went to the mosque in order to calm them. The fighting became less. My dad still doesn't

---

[22]   Interview Omar, 8 December 2016, Casablanca.

[23]   Interview Hicham, 15 December 2016, Rabat.

know that I'm an atheist. I don't want to cause any more problems. And I will keep it for myself until I'm financially independent".[24]

If the "success" of online activism is measured in the efficiency to mobilize widespread participation in a collective activity that receives public recognition and thus brings change (Shayan 2016), the online jokes of Moroccan non-believers might be considered unsuccessful. However, while sharing jokes in Facebook groups might not be enough, forms of online and offline activism can also nourish each other. For example, reading jokes about breaking the fast during Ramadan online, might encourage people to eat in public. This offline–online cycle can also go the other way around, as activism online does not take place in an isolated and neutral space, but is closely connected to the local, national, and international context. In particular, "reaction memes" respond to current news, events, or other happenings (Aguilar et al. 2017). For instance, as soon as someone gets arrested for eating in public, many posts fill the timelines. The added value of offline protest should not be underestimated. Face-to-face interactions of telling a joke are needed to move beyond virtual-only ties and to sustain change in the long term (Shayan 2016). In addition to that, it should not be forgotten that behind jokes on the Internet are real people that put physical, psychological, and emotional effort into online activism.

The few jokes that do reach the Muslim majority provoke mixed reactions. As we have seen based on the numerous examples, some jokes, in order to challenge religious authority and to receive attention, can be quite bold and sometimes also apply vulgar jargon. As jokes are often exaggerated to make them funnier, they often lack the nuance of a normal conversation. This can cause offense and polarization. Humor can thus hurt the subject of the joke or the group that feels addressed, especially when stereotypes about Muslims are reinforced. Thus, while non-believers face clichés themselves, they also engage in reproducing stereotypes, such as portraying believers as blindly following their beliefs. Hence, a mutual stereotyping takes place that rather increases differentiation than that it fosters understanding. Other jokes amplify oppositions, such as violence and peace or belief and science.

Within both the group of non-believers and the Muslim community, huge differences in what is considered "funny" exist. For example, many but not all Muslims might find it hurtful to see a cartoon about the prophet Muhammad. At the same time, many but not all non-believers claim their right to make jokes about religion. Thus, some activists consciously test the borders in their pursuit of freedom of expression and argue that blasphemy laws only work to the disadvantage of minorities, such as non-believers. For many, joking about religion is not necessarily blasphemy, but rather a creative and unrestrained critique. According to Salim, an active member of the civil society, "It does not matter if you agree with these (sometimes extreme) statements or not, it is about the right to say whatever you want."[25] Furthermore, Moroccan-born co-founder of the MALI movement Zineb El Rhazoui became a subject of this discussion, as she was a former journalist at Charlie Hebdo and often vehemently defended cartoons about Islam.

Others are more critical about the unlimited defense of blasphemous jokes and point at the overemphasis on Islam in jokes and biased understanding of putting freedom of speech higher than religious offense. They criticize the discourse, in which religious criticism is associated with freedom and rationality, whereas religious censure is often linked to intolerance, arbitrary, and coercion (Asad et al. 2013). As we deal with a Muslim majority context, it is important not to apply a Western-Christian lens on religious-related jokes. As many scholars have pointed out, Islam has a different relationship to blasphemy. According to Mahmood (in Asad et al. 2013), for many Muslims, the connection to the prophet can be very close and personal and signs can be real embodiments. Therefore, many Muslims might perceive the negative iconography in cartoons or jokes about the prophet as a direct assault and moral injury.

---

24    Interview Soufiane, 18 December 2016, Rabat.

25    Interview Salim, 7 May 2019, Rabat.

In short, while for some, humor should be without limits, others insist on borders that must not disrespect other people's beliefs. Where we draw the line between artistic freedom, religious criticism, and blasphemy is conditioned by the personal and religious values of each person, as well as the context in which the joke is made. Taking both positions into account, it is useful to reflect once more on the punch theory. While in the French debate following the Charlie Hebdo attacks, it was mainly argued that satire is there to punch in all directions (up and down), in other contexts, satire is mainly used to punch up. Thus, usually, satire is a critique of the powerful made by the disempowered. Who is the one in power depends very much on the context and the specific positionality. While in Europe Muslims, are part of a discriminated minority, in Morocco, Islam is the religion of the majority and closely linked to power. Therefore, punching towards Islam has different meanings in both contexts. This proves again that humor is relative (Driessen 2015): in one context, a cartoon about religion can be a tool of expressing liberty, and in another, it can be a visualization of intolerance towards Islam or other religions. In other words, "one's sense of humor may be another's offense" (Molokotos-Liederman 2019).

## 6. Conclusions

Coming back to the initial question of to what extent humor is a tool for Moroccan non-believers to challenge the religious status quo, the following conclusions can be drawn: the numerous cartoons and memes shared in the four Facebook groups that are especially popular among young, urban Moroccan non-believers, shows that humor plays a crucial role. Jokes can give an indication about which topics are important to non-believers. These subjects include direct criticism about the legal restrictions of liberties for non-believers and perceived allies such as the LGBTQ+ community and feminist groups. Other jokes mock religion or reflect on society's views towards non-believers. Not every joke carries an activist message or is told with a political intention, yet, many jokes have the potential to convey controversial opinions that challenge religious expectations and questions of authority.

Humor can therefore be seen as a form of resistance. Jokes on the Internet offer a way to covertly express dissent in a country where religious authority is perceived as hegemonic and closely linked to political legitimacy. In this context, non-believers perceive the Internet as a relatively safe space, while being aware of perils, such as surveillance. Jokes can break the taboos about leaving and criticizing religion and make the unspeakable discussable. Hereby, most jokes do not target a single person that incorporates religious power, such as the King, but rather criticize religious authority as a structural component that permeates the whole society. Nevertheless, joking about religion and its messengers, as well as religious leaders and followers, can undermine the religious basis of the political system. In addition to that, self-satire indirectly questions the religious status quo by reflecting on the position and personal experiences of being not religious in a Muslim-majority society. While humor can be a powerful weapon of the weak, it can also cross the boundaries of the acceptable, as the line between religious criticism and blasphemy remains thin. Especially in a digitalized and globalized world, jokes can reach others within no time, who might disagree about the funniness of the message. Humor that criticizes religion does not only depend on the context, but also on the (power) position of the person who is telling the joke.

To conclude, challenging authority is one, but not the only purpose. Humor can also bridge incongruities when laughing together and it can lead to a feeling of identification. This becomes especially visible during Ramadan when the social pressure on practicing religion increases. In this period, jokes are a possibility to vent about shared struggles, such as having to pretend to fast. Lastly, humor can also function as differentiation, as has been the case during the COVID-19 pandemic. Most of the jokes contrasted the reactions of religious leaders with those of scientists and doctors. While self-expression, identification, and differentiation might not challenge religious authority directly, they might contribute to activism in the long run. Online communities, such as (closed) Facebook groups can

have different functions. They are not only places to laugh but also to exchange ideas, to mobilize, and to support each other. This is important, as activism does not start out of nowhere, but often begins with informal online debates about social and political issues. Facebook groups, such as "Atheists in Morocco", do not constitute a big movement so far, but mocking religion and humorously reflecting on the situation of non-believers can contribute to vivid discussions that might develop into louder activism in the future.

**Funding:** This research was funded by Marie Skłodowska-Curie Actions, ITN-MIDA 813547.

**Institutional Review Board Statement:** The study was conducted according to the guidelines of the Declaration of Helsinki, and approved by the Ethics Committee of Humanities of the Radboud University (protocol code 2020-5876, approved 09.03.2020).

**Informed Consent Statement:** Informed consent was obtained from all interviewees involved in the study.

**Acknowledgments:** I would like to sincerely thank Karin van Nieuwkerk, Araceli González Vázquez, and Sebastian Elsässer, as well as the two anonymous reviewers for their insightful comments. I would also like to express my gratitude to Rayane Al-Rammal, Josias Tembo, and Hayat Douhan for critically thinking with me. Lastly, I'm grateful to my interviewees for kindly sharing their thoughts and experiences on this topic.

**Conflicts of Interest:** The author declares no conflict of interest.

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
