# Peer review of "Laughing about Religious Authority—But Not Too Loud"

_religions, doi:10.3390/rel12020073_

Round 1

Reviewer 1 Report

Lines 113-118 (need corroboration/citation)

The topic is interesting. Regarding data analysis, in the "content of jokes" section, I probably would also expect to see how jokes in atheistic circles differ from ones in the others and so on... This might help to measure the extent of "not too loud" of each group and its revolutionary potential on a graph. Of course, that would require choosing some variables for measurement.

Author Response

Thank you very much for your review. I like the idea to make a graph but I'm not sure how feasible it is in the context of this qualitative study. Could you maybe elaborate a bit more on how you imagined this graph? What would be the variables in question (the challenging nature of the joke?) and between which atheistic circles (the four 4 Facebook groups?)? Maybe it would be more feasible to provide percentages about the topics mentioned in the different groups, e.g. group A 1) criticizing the legal situation 20%, 2) mocking religion 60%, and 3) reflecting on society's views towards non-believers 20%. But also this might be difficult to accomplish as the topics can be overlapping, e.g. one joke can be self-reflecting and critical about religion and society. I would be curious to hear back from you on this point. For now, I added a paragraph (line 22-234) in the content section describing the different trends visible in the four groups. 

Reviewer 2 Report

This is a fascinating article, opening a new window on the role of humour in religion and among those who are 'none'. Humour in this case is directed toward authority, but is respectful and genuinely amusing.

The setting for this study is carefully described which is very important in understanding what is going on and to ensure that any cross cultural comparisons can be made, but only if similarly nuanced in contextual description.

This article is carefully grounded in the literature and adds substantially to the social analysis of humour, religious authority and religious/worldview diversity in Morocco. 

a well written presentation of the humour discovered is couched in a helpful categorisation of humour and followed by a critical analysis of the limits of this form of humour as a tool of activism.

Author Response

Thank you for this positive and encouraging review. Indeed, I hope that this article can contribute to the study of the role of humor in relation to religion, with respect to those that identify as not religious and aim to challenge religious authority. I'm happy to hear that you consider the context of the study as well described and carefully grounded in the literature.